# Harmony4D: A Video Dataset for In-The-Wild Close Human Interactions

**Rawal Khirodkar**\*, **Jyun-Ting Song**\*, **Jinkun Cao, Zhengyi Luo, Kris Kitani**
Carnegie Mellon University

## Abstract

Understanding how humans interact with each other is key to building realistic multi-human virtual reality systems. This area remains relatively unexplored due to the lack of large-scale datasets. Recent datasets focusing on this issue mainly consist of activities captured entirely in controlled indoor environments with choreographed actions, significantly affecting their diversity. To address this, we introduce Harmony4D, a multi-view video dataset for human-human interaction featuring in-the-wild activities such as wrestling, dancing, MMA, and more. We use a flexible multi-view capture system to record these dynamic activities and provide annotations for human detection, tracking, 2D/3D pose estimation, and mesh recovery for closely interacting subjects. We propose a novel markerless algorithm to track 3D human poses in severe occlusion and close interaction to obtain our annotations with minimal manual intervention. Harmony4D consists of 1.66 million images and 3.32 million human instances from more than 20 synchronized cameras with 208 video sequences spanning diverse environments and 24 unique subjects. We rigorously evaluate existing state-of-the-art methods for mesh recovery and highlight their significant limitations in modeling close interaction scenarios. Additionally, we fine-tune a pre-trained HMR2.0 model on Harmony4D and demonstrate an improved performance of 54.8% PVE in scenes with severe occlusion and contact. Code and data are available at https://jyuntins.github.io/harmony4d/.

"*Harmony*—a cohesive alignment of human behaviors."

## 1 Introduction

As social beings, humans frequently interact with each other using physical touch [73, 64, 35]. By studying these interactions, one can potentially unravel various aspects of human behavior, including emotions [24], intentions [13], and dynamics [59]. As with most problems in computer vision [36], a first step in modeling contact interactions involves building large-scale 3D multi-human datasets. Many such datasets [15, 74, 20, 80, 87, 57] have emerged in recent years. However, similar to most existing single-human datasets [28], contact interaction datasets lack subject and environment diversity and are captured under controlled indoor conditions with choreographed activities. Learning-based methods [42, 70, 43, 38] trained on such biased benchmarks struggle to generalize to real-world conditions. The core issue is that recovering high-quality ground-truth mesh for scenarios with frequent human-human contact is challenging due to severe occlusion, truncation, and dynamic movements [58]. Existing methods typically rely on extensive RGBD motion capture systems [80] or a large number of high-end wired camera systems (over 100) [31] to achieve accurate annotations. This reliance on extensive static capture systems makes in-the-wild data collection impractical [60]. Therefore, we ask: can we develop a markerless capture system that uses only a few cameras, is mobile, and is capable of accurately extracting 3D ground truth for in-the-wild scenarios involving contact interactions? To tackle this challenge, we introduce the *Harmony4D* dataset.

---

\*Equal contribution

38th Conference on Neural Information Processing Systems (NeurIPS 2024) Track on Datasets and Benchmarks.

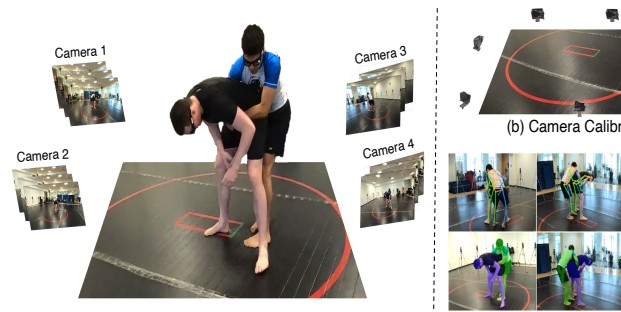
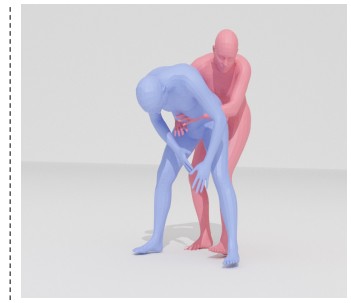

Figure 1: **Overview of Harmony4D setup**. (a) Multiple synchronized and calibrated cameras capture the contact interaction in *wrestling*. (b) We align all cameras into a gravity-aligned metric world coordinate system. (c) Our processing obtains per-view instance segmentation masks along with 3D keypoints. (d) Reconstructed ground-truth meshes after multi-step collision optimization.

Harmony4D is a novel dataset featuring high-resolution videos of dynamic activities with contact interactions such as wrestling, dancing, karate, MMA and fencing. In contrast to previous datasets, Harmony4D is collected in the wild with a specific focus on subject and environment diversity. Table 1 compares our dataset with existing 3D human datasets focusing on contact interactions. Harmony4D is a substantially large dataset, consisting of 1.66 million images captured from more than 20 synchronized cameras, resulting in 3.32 million visible human instances. Specifically, we provide comprehensive ground-truth annotations such as camera parameters, 2D bounding boxes [23], human tracking identities [86], 2D/3D human poses [68], and 3D human meshes [33]. Figure 1 provides an overview of the capture setup and annotation processing. The multi-camera setup is inspired by EgoHumans [39], utilizing Meta's Aria glasses [54], which feature an RGB and two greyscale cameras for the subject's first-person view, along with stationary RGB cameras for the third-person view. This combination allows us to accurately track and triangulate poses in 3D for extended periods without using visual markers [85] or additional sensors [1]. To our knowledge, Harmony4D is the only in-the-wild video dataset with dynamic activities and contact interactions.

Our annotation procedure minimizes the need for manual supervision. We divide any input multi-view video sequence into two stages: (i) pre-contact and (ii) post-contact. The pre-contact stage refers to the time interval before the first physical interaction between the subjects. We utilize an existing pose extraction algorithm [39] to obtain 3D poses during the pre-contact stage. However, existing methods face significant challenges in post-contact scenarios, primarily due to severe occlusion, truncation, and joint ambiguity when subjects are in very close proximity (e.g., during wrestling or dancing). For the challenging post-contact stage, we propose a novel algorithm that uses instance segmentation [40], segmentation-conditioned 2D pose estimation [51], and 3D pose forecasting [2] in a temporal feedback loop to accurately track 3D poses. Our key idea is to use segmentation-conditioned 2D pose estimation, see (c) in Figure 1, to reason about missing or completely hidden body parts and disambiguate between multiple human joints. Finally, we build an efficient multi-stage motion capture pipeline to fit the SMPL [52] body model to the 3D human skeletons, incorporating optimization to minimize mesh interpenetration.

| Dataset | In-The-Wild | Scenes | Subjects | Cameras | Images | Instances | Mesh |
|---|---|---|---|---|---|---|---|
| ShakeFive2 [74] | ✗ | 1 | 6 | 1 | 34K | 68K | ✗ |
| MuPoTs-3D [57] | ✗ | 3 | 8 | 8 | 8K | 22K | ✗ |
| MultiHuman [87] | ✗ | 1 | 8 | 6 | 32K | 69K | ✓ |
| ExPI [20] | ✗ | 1 | 4 | 68 | 1.9M | 3.8M | ✗ |
| CHI3D [15] | ✗ | 1 | 10 | 4 | 315K | 728K | ✓ |
| Hi4D [80] | ✗ | 1 | 40 | 8 | 88K | 176K | ✓ |
| **Harmony4D (Ours)** | ✓ | 5 | 24 | 20 | 1.66M | 3.32M | ✓ |

Table 1: **Comparison with existing 3D datasets with multi-human interactions.** *Subjects* and *Scenes* are number of unique subjects and capture environments. *Cameras* are number of stationary views per sequence. *Images* are number of images. *Instances* are number of visible human instances.

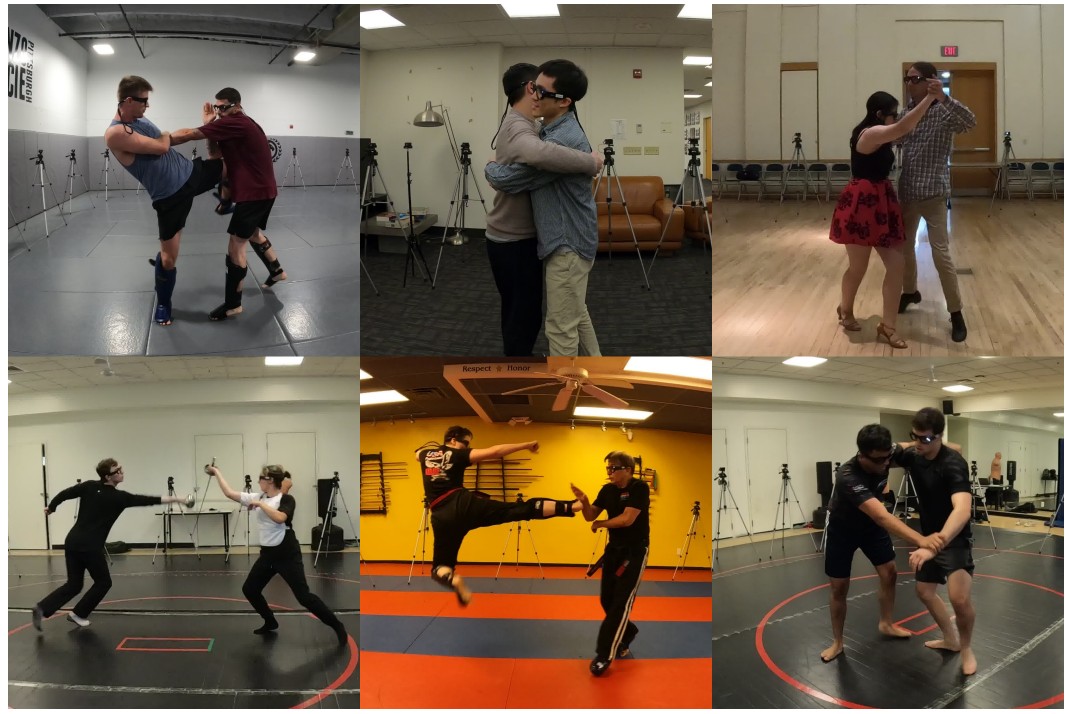

Figure 2: **Dataset Composition.** Harmony4D consists of diverse, dynamic activities such as dancing, karate, MMA, and wrestling, all captured in in-the-wild settings.

The extensive scale and diverse scenarios of the Harmony4D dataset enable a thorough evaluation and improvement of methods for human-human contact estimation. We specifically evaluate current techniques for human mesh regression, uncovering that existing methods often struggle with missing meshes, inaccurate pose predictions in the presence of occlusion, and handling the complexities of natural, unconstrained human interactions. Importantly, when we fine-tune off-the-shelf methods on our large training set, the fine-tuned methods generalize well to challenging contact interactions and even outperform specifically designed methods for human contact reasoning [58]. Moreover, we observe significant improvements in vertex contact prediction and occlusion reasoning. This underscores the need for Harmony4D, a large-scale dataset and a robust evaluation benchmark for in-the-wild contact interactions. The limitation is not necessarily with the methods [42, 70, 48, 46], but with the need to expose these methods to more extensive data in these underrepresented scenarios.

Our contributions are summarized as follows.

- A novel method based on multi-view instance segmentation and 3D human pose forecasting to extract 4D meshes of closely interacting humans.
- Harmony4D, a large-scale in-the-wild dataset with millions of multi-view images, parametric body models and vertex-level contact for dynamic and unchoreographed activities.
- Evaluation of existing state-of-the-art methods for monocular mesh regression, emphasizing their fundamental limitations in handling contact interactions, and demonstrating significantly improved performance when fine-tuned on our dataset.

## 2  Related Work

**Limited 3D Mesh Recovery Datasets.** Current progress in human vision research has been significantly driven by datasets [11, 76, 51, 28, 10, 16, 25, 45, 84]. However, unlike 2D pose datasets, 3D human mesh estimation datasets [28, 75, 56, 31, 32, 67, 79] are limited in diversity which significantly hampers the ability of deep models to generalize to the real world [82]. Popular 3D datasets like Human3.6M [28], AMASS [55], HumanEva [67], AIST++ [47], HUMBI [81], PROX [22], and To-talCapture [32] only contain single human sequences. Multi-human datasets like PanopticStudio [31],

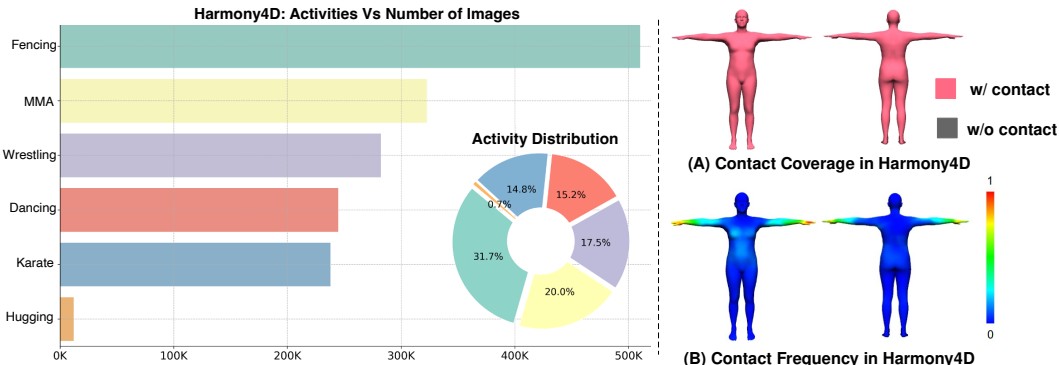

Figure 3: **Data Distribution.** The dynamic activities in the Harmony4D dataset cover all area for the SMPL body model. We visualize the most frequent body parts in contact during interactions as a normalized heatmap.

MuCo-3DHP [57], TUM Shelf [8] are limited to indoor lab conditions. Outdoor multi-human datasets like 3DPW [75], MuPoTS [57], and EgoHumans [39] do not focus on human-human interactions. The Harmony4D dataset goes beyond existing 3D mesh datasets in meaningful ways, capturing in-the-wild activities with frequent multi-human contact and a particular focus on subject and scene diversity, as well as unchoreographed activities.

**Human Pose and Shape Estimation.** Most approaches [49, 42, 50, 46, 19, 12, 83, 33, 9, 38, 69] rely on the SMPL model [52], which provides a low-dimensional parametrization of the human body. HMR [34] employs a neural network to regress the parameters of an SMPL body model from a single image. Follow-up works like 4DHumans [18], Multi-HMR [3], WHAM [66], BEV [70], ROMP [69], and PARE [42] have improved the robustness of the original method by using more annotations, larger models, and auxiliary conditioning information such as camera parameters, segmentation masks, and 2D poses. However, most methods require "full-body" single human images [62], limiting their robustness in scenarios where multiple humans are interacting due to the limited representation in the training data. Recent works like BUDDI [58] build a diffusion model on top of BEV [70]'s output as initialization to model the distribution of humans in proximity. Despite these advancements, we show that existing methods fail in scenarios with interacting humans in the Harmony4D dataset.

**Close Human Interaction Datasets.** Several contact-related datasets focus on human interactions with objects or static scenes [72, 4, 14, 22, 27, 71]. Additionally, recent datasets model close interactions between dynamic humans [80, 15, 74, 26, 31, 57, 20, 87]. ShakeFive2 [74] and MuPoTS-3D [57] only provide 3D keypoints and lack mesh or body shape information. CHI3D [15] uses an indoor motion capture system to fit parametric human models of at most one actor at a time. MultiHuman [87] provides textured scans of interacting people but lacks ground-truth level body model registrations. ExPI [20] contains dynamic textured meshes in addition to 3D joint locations but misses body model registrations and contact information. Furthermore, ExPI [20] includes only two pairs of dance actors. The most related dataset to ours is Hi4D [80], which uses a multi-view RGBD capture system to capture the 4D volume of two subjects interacting with each other. Hi4D uses online tracking and optimization to extract per person scans from the joint scans and fit a parametric body model to them. However, Hi4D is limited to a single indoor capture location and lacks background diversity, consisting of choreographed activities. In contrast, Harmony4D is collected in-the-wild settings with unchoreographed dynamic activities.

## 3   Harmony4D

This section describes the data collection setup and our proposed in-contact human mesh tracking system. Our objective is to develop a markerless annotation pipeline that accurately provides ground truth 3D human shapes and poses from videos, even in cases of severe occlusions and multiple human contacts, with minimal manual intervention. The proposed capture and mesh tracking system builds upon EgoHumans [39] and extends to work effectively under post-contact conditions, including significant occlusions and truncations.

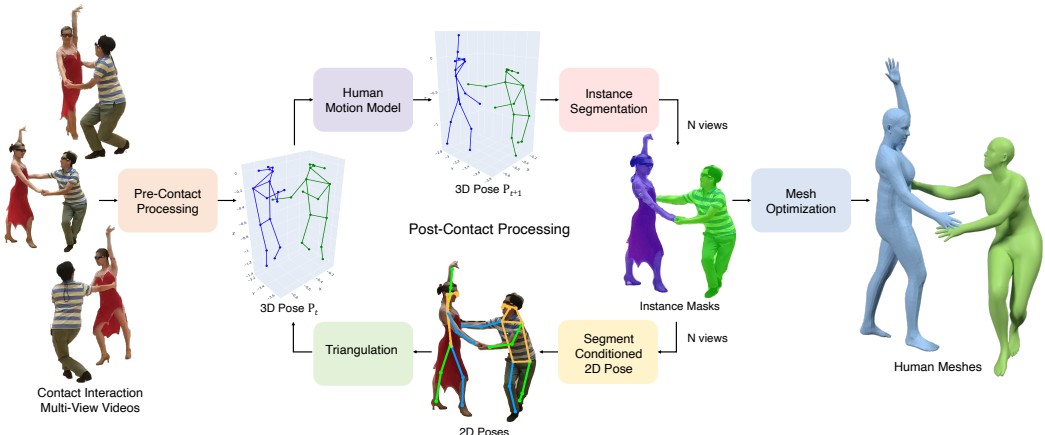

Figure 4: **Overview of Harmony4D processing setup**. Given a multi-view RGB video sequence, we divide it into pre-contact and post-contact stages. We estimate per-subject 3D poses in the pre-contact stage [39] as initialization. The post-contact stage uses sequential processing involving 3D pose forecasting with a human motion model, per-view 2D point-conditioned instance segmentation, and mask-conditioned 2D pose estimation, followed by multi-view triangulation and mesh fitting.

**Data Collection.** We aim to capture dynamic activities with human-human contact under in-the-wild conditions such as wrestling, dancing, fencing, etc. Figure 2 and Figure 3 show the captured dynamic activities and depicts the data distributions of activities in the Harmony4D dataset. In comparison to previous datasets, our sequences are not restricted to indoor conditions and consist of realistic contact interactions with over $1.6$ million images. Following EgoHumans [39], to obtain high-quality ground truth, our capture setup includes multiple views using 20 GoPro cameras, refer Figure 1 (b). The video resolution is set to 4K ($3840 \times 2160$) and recorded at a rate of 60 frames per second (FPS). Importantly, the volume created by our cameras is portable and can be moved across locations. All cameras are synchronized to ensure temporal consistency across different views. Optionally, we also include Aria glasses [54] to provide the egocentric perspective of subjects. Each sequence consists of two subjects. All participants were briefed on the research project, provided informed consent following IRB guidelines, and received monetary compensation for their participation.

**Camera Calibration.** We determine the intrinsic and extrinsic parameters for all cameras using structure from motion (SfM) [65] for each sequence. The world coordinate system is scaled to be metric and gravity-aligned. To ensure the registration of all cameras in a single coordinate system using SfM, we pre-scan the environment externally with an additional camera. For contact sequences with Aria glasses [54], we also transform the camera coordinates of the ego-glasses into the stationary camera coordinate system using Procrustes alignment [53].

**Pre-Contact Processing.** We divide a multi-view video sequence into two parts: (i) pre-contact, the time interval before the first subject-to-subject contact, and (ii) post-contact, the period after the first contact. We leverage the multi-person 3D pose estimation method from EgoHumans [39] to obtain 3D poses in the pre-contact stage. Our pre-contact processing is efficient, parallelizing all time-steps, and works accurately since the subjects are completely visible from most camera views during this stage.

### 3.1 Post-Contact Processing.

The main challenges in post-contact are detecting partially or completely occluded keypoints, associating these keypoints with the correct human identities, and ensuring that the estimated 3D human meshes remain spatially and temporally coherent while being consistent with the multi-view video evidence. To address this, we propose a novel sequential algorithm that leverages 3D human-pose forecasting, 2D point-conditioned instance segmentation, and mask-conditioned 2D pose estimation. Figure 4 provides an overview of Harmony4D post-contact processing.

**Human Motion Model.** To reason about occluded keypoints, we use 3D human pose forecasting with a human motion model based on Kalman filter (KF) [44]. KF is a linear estimator for dynamical systems discretized in the time domain which only requires the state estimations on a history of time

steps to estimate the next time step target state. Specifically, we train a per-subject 3D motion model in a sliding window fashion over a history of $T$ frames to predict the future 3D keypoint locations. The forecasting model is initialized using the pre-contact poses. Assuming $J$ keypoints, we use $J$ filters to model the 3D motion of each keypoint independently. Each KF's state $\mathbf{x}$ is defined as $\mathbf{x} = [x, y, z, \dot{x}, \dot{y}, \dot{z}]^\top$, where $(x, y, z)$ are the 3D keypoint coordinates and $(\dot{x}, \dot{y}, \dot{z})$ are the velocities.

**Instance Segmentation.** We aim to infer the actual 3D keypoint locations at time step $t + 1$ from the forecasted ones, accounting for discrepancies due to sudden motions like tackling or jumping. Most current methods rely on per-view 2D pose estimation using a bounding box and then multi-view triangulation [29]. However, bounding boxes are ambiguous [37] when subjects are close together, as seen in Figure 5 (*Left*). An off-the-shelf 2D pose estimator, given the subject bounding-boxes is unable to distentangle the subject poses accurately. Our key idea is to use instance segmentation masks as conditioning to a 2D pose estimator. We project the forecasted 3D poses to all views and apply the Segment-Anything model [40] using the target subject's 2D pose as positive points and others' poses as negative points. Our high frame rate processing ensures the forecasted 3D poses are close to the true ones, enabling reliable instance mask estimation across all views.

**Segmentation Conditioned 2D Pose Estimation.** We propose SegPose2D, a deep model for conditional 2D pose estimation, which takes both the RGB image patch and the binary segmentation mask as input to predict the 2D pose. SegPose2D uses the ViTPose [78] backbone with two transformer branches and feature fusion at multiple depths of the network. We train SegPose2D on the COCO [51] dataset using ground-truth segmentation masks. Figure 5 (*Right*) compares the mask-conditioned 2D pose estimation of SegPose2D with ViTPose [78] on a hugging sequence. The input mask is crucial for disentangling occluded human poses.

**Multi-View Triangulation.** Our triangulation setup follows the pre-contact processing [39]. Let $C$ represent all synchronized video streams with known projection matrices $P_c$. We aim at estimating the global 3D pose $\mathbf{y}_{j,t} \in \mathbb{R}^3$ of a fixed set of human keypoints indexed by $j \in (1..J)$ at timestamp $t \in (1..T)$ for all humans in the scene (omitting the human index for simplicity). Let $\mathbf{x}_{c,j,t} \in \mathbb{R}^2$ be the $j$th 2D keypoint at time $t$ from camera $c$. To infer 3D poses from 2D estimates, we use a linear algebraic multi-view triangulation approach [21]. Traditional triangulation assumes equal contribution from all 2D keypoints $\mathbf{x}_{c,j,t}$, but some views may be unreliable due to occlusions or framing issues, degrading the results. We apply RANSAC to address this. For each time step $t$, we solve: $\tilde{\mathbf{y}}_{j,t}, A_{j,t}\tilde{\mathbf{y}}_{j,t} = 0$, where $A_{j,t} \in \mathbb{R}^{2C' \times 4}$ consists of components from the projection matrices and $\mathbf{x}_{c,j,t}$ and $C'$ is the cardinality of the camera inlier set post-RANSAC. Finally, we refine the 3D poses for the entire sequence using temporal smoothing, joint symmetry and bone constraints [7, 39].

**Mesh Optimization.** Given the 3D pose estimates $\mathbf{y}_{\{1..T\}}$ for all subjects in a video sequence, we fit a human mesh to these 3D pose sequences to obtain the in-contact vertex annotations. The human mesh is represented using body pose and shape parameters, $\boldsymbol{\theta} = [\boldsymbol{\theta}_{\text{pose}}, \boldsymbol{\theta}_{\text{shape}}, \boldsymbol{\theta}_{\text{global}}]$, where $\boldsymbol{\theta}_{\text{pose}} \in \mathbb{R}^{23 \times 6}, \boldsymbol{\theta}_{\text{shape}} \in \mathbb{R}^{10}, \boldsymbol{\theta}_{\text{global}} \in \mathbb{R}^6$. The pose parameters $\boldsymbol{\theta}_{\text{pose}}$ are the 6D representation of the joint rotations [88] of the 23 body joints of the SMPL [52] body. The shape parameters $\boldsymbol{\theta}_{\text{shape}}$ are the first 10 coefficients of the PCA shape space derived from CAESAR [63] scans. $\boldsymbol{\theta}_{\text{global}}$ consists of the global root orientation and translation of the body. Let $\Phi : \boldsymbol{\theta} \to \mathbf{y}$ be a differentiable mapping function that projects SMPL parameters $\boldsymbol{\theta}$ to corresponding 3D keypoints $\mathbf{y}$. Similar to EgoHumans [39], we fit $\boldsymbol{\theta}$ to the 3D pose trajectory by minimizing $\mathcal{L}_{\text{mesh}}$ defined as follows,

$$\mathcal{L}_{\text{mesh}}(\boldsymbol{\theta}) = w_1 ||\mathbf{y} - \Phi(\boldsymbol{\theta})||_2 + w_2 ||\boldsymbol{\theta}_{\text{pose}}||_2 + w_3 \mathcal{L}_{\text{limb}}(\Phi(\boldsymbol{\theta})) + w_4 \mathcal{L}_{\text{symm}}(\Phi(\boldsymbol{\theta}))$$
$$+ w_5 \mathcal{L}_{\text{temporal}}(\Phi(\boldsymbol{\theta})) + w_6 \mathcal{L}_{\beta}(\boldsymbol{\theta}_{\text{shape}}) + w_7 \mathcal{L}_{\text{collison}}(\theta) \quad (1)$$

| Input | Instance Mask A | Instance Mask B | Instance Mask A | ViTPose | SegPose2D (Ours) | Instance Mask B | SegPose2D (Ours) |

Figure 5: (*Left*) Point conditioned instance segmentation. Projected 3D keypoints as positive or negative prompts. (*Right*) Comparison of ViTPose [78] with mask-conditioned 2D pose estimation.

where $\mathcal{L}_{\text{limb}}$ is the constant limb length loss, $\mathcal{L}_{\text{symm}}$ is the body symmetry loss, $\mathcal{L}_{\text{temporal}}$ is temporal smoothing, $\mathcal{L}_{\beta}$ is the Gaussian mixture shape prior loss [6], $||\boldsymbol{\theta}_{\text{pose}}||_2$ penalizes hyper-extensions of joints, $\mathcal{L}_{\text{collison}}$, inspired by Interdiff [77], prevents mesh self and interpenetration and $w_1..w_7$ are scalar weights. Compared to optimization in EgoHumans [39] where each subject mesh is fitted independently, the addition of the $\mathcal{L}_{\text{collison}}$ is a crucial improvement for modeling in-the-wild multi-human contact scenarios. We define $\mathcal{L}_{\text{collison}}(\boldsymbol{\theta}) = \Sigma_{i=1}^{N}\Sigma_{j=1}^{V}\phi_i(x_j, y_j, z_j)$, where $N$ is the number of subjects, and $V$ is the number of mesh

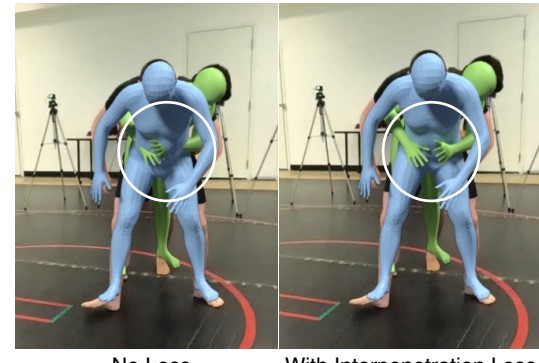

No Loss      With Interpenetration Loss

Figure 6: Mesh optimization with interpenetration loss.

vertices. $\phi$ computes the negative penetration depth by $\phi(x, y, z) = -\min(\text{SDF}(x, y, z), 0)$ where $\text{SDF}(x, y, z)$ is the signed-distance field [30] at a vertex location $(x, y, z)$. This formulation ensures that the loss is positive when vertices penetrate themselves or other human meshes, thereby encouraging the separation of two human meshes as shown in Figure 6.

# 4 Experiments

In this section, we first describe the evaluation of mesh recovery methods on our dataset. Then, we showcase ablative experiments that highlight the quality of the annotations provided by the dataset.

## 4.1 Implementation Details

We divide each sequence into shorter clips of at least 5 seconds at 20 FPS. The annotation per time step includes camera parameters, bounding boxes, person IDs, 2D/3D human poses, and 3D meshes per subject. All 3D poses at each time step are manually inspected and rectified in case of errors before performing mesh fitting. After mesh optimization, we manually verify and re-optimize each mesh sequence and contact vertices with custom hyperparameters if necessary. The number of keypoints $J$ is set to 17 [51]. The human motion model uses history of 10 frames. We use the Segment-Anything-H [40] backbone for instance segmentation. SegPose2D is based of the ViTPose-H [78] backbone and is trained on 6 A100 GPUs for 5 days on the COCO [51] dataset. We use CLIFF [48] to obtain initial SMPL estimates for mesh optimization. To compute the

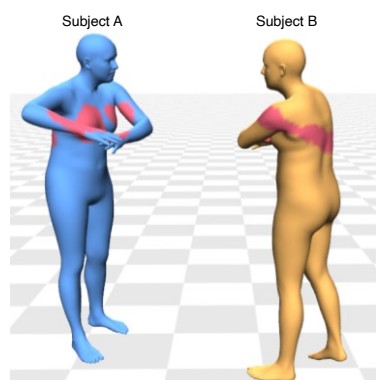

Subject A      Subject B

Figure 7: Ground-truth contact vertices for *hugging* sequence.

SDF for mesh interpenetration loss, we use a custom GPU implementation by Jiang et al. [30] along with Geman-McClure [17] error for robustness. Figure 8 provides a multi-view visualization of the ground-truth mesh during contact interaction. We visualize the vertices in contact in Figure 7. For more visualization, please refer to the supplemental.

## 4.2 Benchmarking Mesh Recovery Methods

We evaluate existing state-of-the-art monocular mesh prediction methods such as PARE[41], HMR2.0[18], ROMP[69], BEV[70], Multi-HMR[3] and BUDDI[58] on the Harmony4D test set. For fairness, we use ground-truth bounding boxes as input to the top-down methods.

**Metrics.** We report standard metrics [75] such as MPJPE and PVE for mean joint and vertex errors, along with PA-MPJPE, PA-PVE, N-MPJPE, and N-PVE for their Procrustes-alignment and F1 score normalized variants, respectively. Additionally, we also report other metrics like 3DPCK, AUC [60]. To measure interpenetration during multi-contact, we compute maximum point-to-surface distance (mP2S) in mm. Note, for BUDDI and Multi-HMR, which predict SMPL-X [61] parameters instead of SMPL, we follow BEDLAM [5] and convert predicted SMPL-X meshes to SMPL using a fixed vertex mapping $\mathcal{M} \in \mathbb{R}^{10475 \times 6890}$.

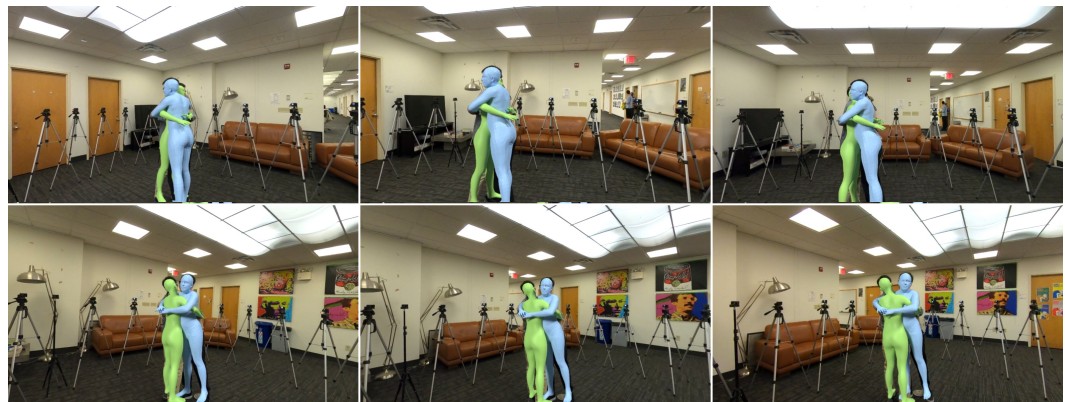

Figure 8: Harmony4D ground-truth mesh visualized from six camera views for the *hugging* sequence.

**Discussion.** In contrast to a general dataset like 3DPW [75], we observe that the MPJPE for baseline methods is much higher on the challenging Harmony4D test set. For instance, off-the-shelf Multi-HMR [3] reports an MPJPE of 61.4 mm on 3DPW compared to 93.8 mm on the Harmony4D benchmark. Notably, Multi-HMR, a bottom-up method, performs significantly well compared to other baselines, even outperforming BUDDI [58], which is specifically trained and designed to model human-human contact. Among the top-down methods, HMR2.0 [18] performs the best with an MPJPE of 108.2 mm due to its large-scale training and the ViTPose [78] transformer backbone. Importantly, we also notice a very high normalized per vertex error (N-PVE) for all baselines, which is due to the failure in predicting a 3D consistent mesh in the presence of frequently occurring occlusions in our dataset. Harmony4D provides a unique and challenging evaluation benchmark for in-the-wild human contact interaction scenarios.

**Finetuning.** To demonstrate the utility of our dataset beyond serving as a challenging evaluation benchmark, we finetune HMR2.0 [18] on our train set. The Harmony4D train set is significantly larger than existing general datasets, containing more than 1.2 million images. We use a training setup similar to HMR2.0 [18] for finetuning. HMR2.0-finetuned demonstrates significant improvement across all metrics, see Table 2. It improves MPJPE by 55.9%, MPVPE by 54.8% and 3DPCK by 21.01%. Interestingly, we do not supervise HMR2.0 specifically with contact vertex annotations; however, we observe that the inter-contact relationships are preserved in the predictions. Figure 9 provides a qualitative comparison on the test set between BUDDI [58], HMR2.0 [18], and HMR2.0-finetuned alongside the ground truth annotations. The results show that our train set can serve as an effective source for adapting existing monocular mesh estimation methods to multi-human interaction settings.

### 4.3 Ablations

**Number of Cameras.** We investigate the impact of the number of cameras on the estimated 3D pose accuracy in our processing. Specifically, we uniformly sample 6 to 18 equidistant cameras from the camera perimeter and calculate the MPJPE between the resulting keypoints and our ground truth 3D keypoints obtained from all 20 cameras. Figure 10 shows the performance trend across various

| Method | MPJPE ↓ | PA-MPJPE ↓ | N-MPJPE ↓ | PVE ↓ | PA-PVE ↓ | N-PVE ↓ | 3DPCK ↑ | AUC ↑ | F1 ↑ | mP2S ↓ |
|---|---|---|---|---|---|---|---|---|---|---|
| PARE[41] | 119.03 | 65.49 | 138.40 | 144.77 | 73.52 | 168.34 | 73.23 | 47.19 | 0.86 | N/A |
| ROMP[69] | 121.13 | 80.23 | 134.59 | 161.12 | 91.82 | 179.02 | 68.47 | 44.70 | 0.90 | N/A |
| BEV[70] | 111.29 | 78.04 | 119.66 | 144.28 | 90.13 | 155.14 | 74.63 | 48.94 | 0.93 | 134 |
| BUDDI[58] | 126.35 | 84.00 | 133.00 | 158.72 | 95.95 | 167.06 | 70.28 | 45.75 | 0.95 | 106 |
| HMR2.0[18] | 108.18 | 60.25 | 109.28 | 131.00 | 67.96 | 132.32 | 75.62 | 49.79 | 0.99 | N/A |
| Multi-HMR[3] | 93.82 | 58.53 | 101.98 | 115.79 | 67.67 | 125.85 | 83.57 | 55.08 | 0.92 | 53 |
| **HMR2.0-finetune** | **47.71** (-46.11) | **32.75** | **48.18** | **59.14** | **38.93** | **59.74** | **96.63** | **75.75** | **0.99** | N/A |

Table 2: Comparison of monocular mesh recovery methods on the Harmony4D test set. mP2S metric is not reported for HMR2.0, PARE, and ROMP, as they predict each human instance in an independent coordinate system. Note, finetuning HMR2.0 improves performance significantly.

| Image | Ground-Truth | BUDDI | HMR2.0 | HMR2.0-Finetuned |

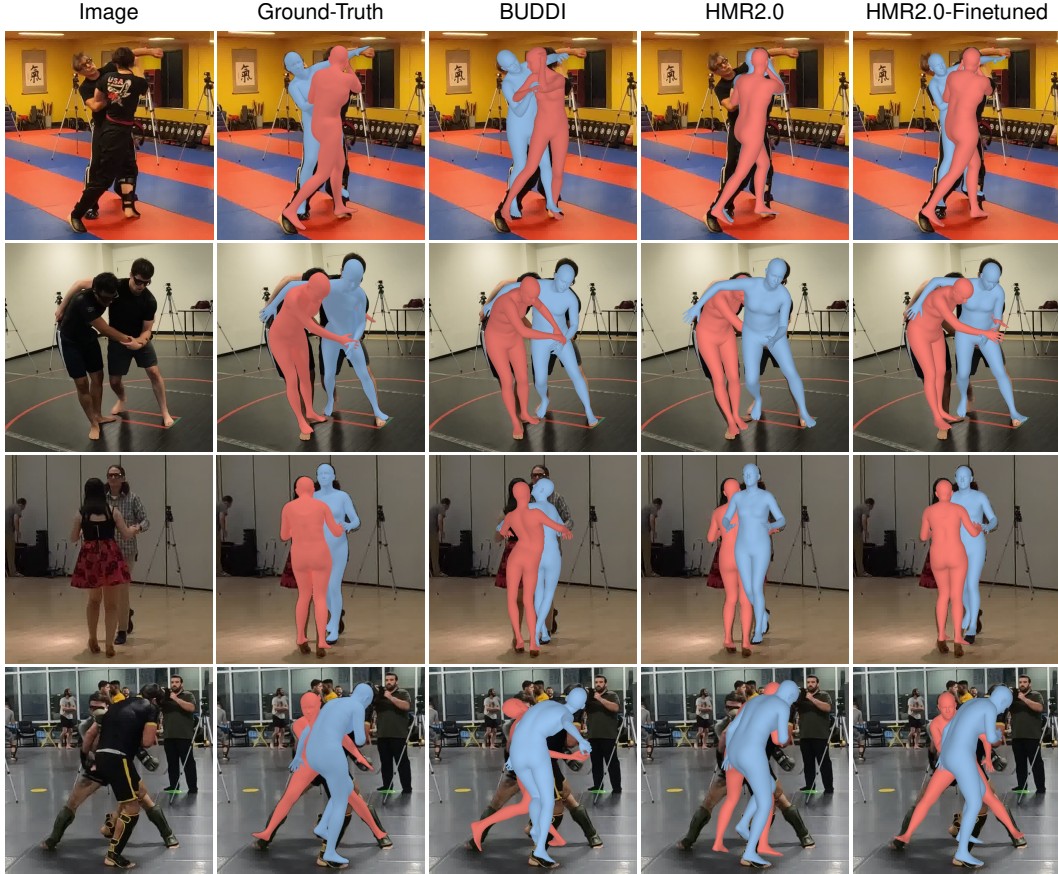

Figure 9: **Qualitative comparison of mesh estimation methods.** We evaluate all methods using ground-truth bounding boxes for fairness. HMR2.0 finetuned on the Harmony4D `train` set, demonstrates robustness to inter-person occlusion and improved 3D in-contact estimation.

activities in our dataset. We observe that increasing the number of cameras predictably improves performance, as more cameras imply a higher likelihood of a larger inlier set during RANSAC in triangulation. Interestingly, 16 cameras offer a fair trade-off between accuracy and processing speed.

**Interpenetration Loss.** We examine the impact of interpenetration loss on our mesh optimization. We evaluate using the following collision metrics: maximum point-to-surface distance (mP2S) in mm, maximum volumetric IoU (mIoU), and maximum intersection of area (mIoA) in $m^2$. Table 3 demonstrates a significant improvement in all collision metrics across sequences. Specifically, on average, mP2S is reduced by $44.7$ mm, mIoU is reduced by $0.7\%$, and mIoA is reduced by $0.1\ m^2$.

| Event | No Loss | | | Interpenetration Loss | | |
|---|---|---|---|---|---|---|
| | mP2S ↓ | mIoU ↓ | mIoA ↓ | mP2S ↓ | mIoU ↓ | mIoA ↓ |
| Hugging | 111.14 | 1.59 | 0.24 | 51.50 | 0.40 | 0.07 |
| Wrestling | 102.88 | 0.79 | 0.16 | 95.00 | 0.61 | 0.10 |
| Dancing | 114.91 | 1.41 | 0.18 | 65.60 | 0.79 | 0.12 |
| Karate | 113.70 | 0.94 | 0.16 | 44.97 | 0.25 | 0.06 |
| MMA | 122.01 | 2.26 | 0.31 | 83.99 | 1.43 | 0.18 |

Table 3: Adopting the interpenetration loss in mesh optimization significantly improves multi-human collision metrics across activities in the Harmony4D dataset.

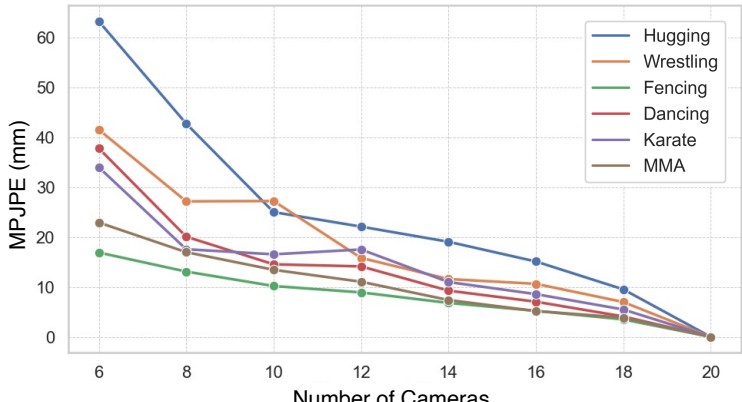

Figure 10: 3D pose triangulation error with varying number of cameras for various interaction activities.

# 5 Conclusion

We propose a novel method to track, segment, and localize 4D body meshes of multiple people interacting in close range with frequent dynamic physical contact under in-the-wild conditions. Our key idea is to use multi-view segmentation-conditioned pose estimation, 3D motion models for forecasting, and collision optimization to obtain precise body model parameters. Using this method, we constructed the diverse Harmony4D dataset with ground-truth annotations for mesh recovery. Emphasis is placed on capturing unchoreographed, dynamic activities such as wrestling, dancing, karate, and MMA in the real world. Our evaluations show that existing state-of-the-art methods fail significantly under the challenging sequences of our dataset, mainly due to the lack of representation of human-human contact interactions in training. Importantly, fine-tuning baselines on our large training set improves mesh estimation performance in severe occlusion and contact conditions.

**Limitations.** We currently trade off the accuracy of 3D pose and shape estimation under contact interactions in favor of capturing in the wild. By adopting an optimization perspective from a dense and mobile camera rig, we enable markerless large-scale capture, complementing high-precision static indoor wired 3D capture systems with limited diversity.

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
