# Harmony4D: A Video Dataset for In-The-Wild Close Human Interactions
# – Supplementary Materials –

**Rawal Khirodkar**[*], **Jyun-Ting Song**[*], **Jinkun Cao,** **Zhengyi Luo,** **Kris Kitani**
Carnegie Mellon University

## 1 Harmony4D

For video demos of Harmony4D, please visit: Harmony4D Website. To download the dataset, please go to download link. Please do not share the dataset with anyone as it is not publicly available yet.

### 1.1 Dataset statistics

Harmony4D is a 75-minute video dataset collected using over 20 eqidistant, synchronized GoPro cameras. It consists of 1.66M images and 3.32M human instances, divided into 1.28M images for training and 376K images for testing, with minimal overlap of subjects between the sets. We manually clipped the videos into 208 sequences across 6 different activities, ensuring each sequence is at least 5 seconds (100 frames) long for temporal continuity. Figure 1 shows images captured by our multi-view system. For details on data distribution across different activities, please refer to Figure 2 in the main paper.

### 1.2 Contents

We released the intrinsic and extrinsic parameters for all cameras, obtained through structure from motion (SfM) [15]. For each image in our dataset, we provide 2D bounding boxes (bboxes), 2D/3D human poses, and human meshes, all along with human IDs. The 2D bboxes are derived from projected SMPL human vertices. The 2D/3D human poses are in both COCO[11] and SMPL[12] formats. The COCO format poses come from our post-contact processing, while the SMPL format poses are derived from the mesh fitting process and regressed using [14]. Our annotations for human meshes include the parameters for SMPL poses, shape, and camera translation, as well as their corresponding vertices. Examples from Harmony4D Dataset are shown in Fig 2. For more qualitative results, please visit our website.

### 1.3 Ego Views

As a by-product of our capture system, we provide annotations for the ego-centric images captured by our Meta's Aria glasses in hugging, fencing, and dancing sequences. We believe these extra annotations will help in understanding close human interactions better when used combined with the exoviews. Figures 3 present our qualitative results from egocentric views.

## 2 Implementation Details

The code used for constructing Harmony4D and evaluation is available here.

---

[*]Equal contribution

38th Conference on Neural Information Processing Systems (NeurIPS 2024) Track on Datasets and Benchmarks.

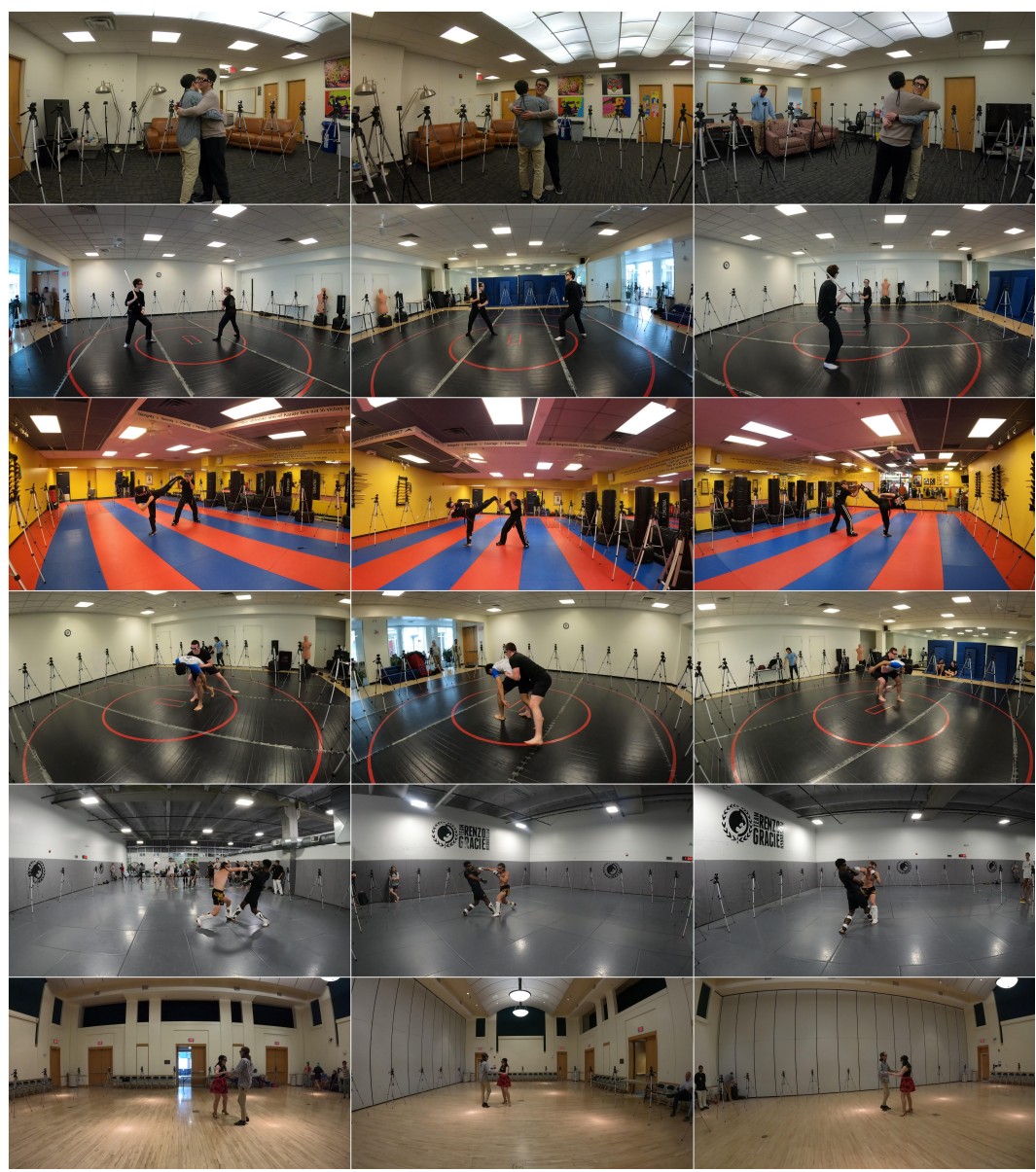

Figure 1: **Capturing human poses from multiple viewpoints.** Our multi-view system captures subjects from various angles, ensuring that human poses blocked from one viewpoint can be seen from another.

## 2.1 Processing Pipeline

**Pre-Contact Processing**   Following [7], for the initial frames before any contact happens, we use HRNet-wholebody[16] to extract 2D keypoints from each view. To ensure these 2D keypoints represent distinct individuals, we compute their object keypoint similarity[11] and filter out potentially duplicated ones using a threshold of 0.7. During triangulation, for each keypoint, we create possible view sets by selecting any two views with keypoint confidence greater than 0.5. Views with a reprojection error below 0.01 are added to these sets. The set with the highest number of views is then selected as the inlier set for that keypoint.

**Post-Contact Processing**   We use a sliding window of size 10 to build our human motion model. During the sequential processing, the 2D keypoints detected by our proposed segpose2D model are refined using a confidence threshold of 0.2. The subsequent triangulation step is the same as in Pre-Contact Processing. Following [7], we further refine the 3D pose estimates $y_{\{1..T\}}$ globally by

| Stage | $w_{joint}$ | $w_{reg}$ | $w_{limb}$ | $w_{symm}$ | $w_{temporal}$ | $w_\beta$ | $w_{collision}$ |
|---|---|---|---|---|---|---|---|
| 1 | 1.0 | 0.0 | 0.0 | 0.0 | 0.0 | 0.0 | 0.0 |
| 2 | 0.0 | 0.0 | 1.0 | 0.2 | 0.0 | 0.005 | 0.0 |
| 3 | 10.0 | 0.001 | 1.0 | 0.0 | 1.0 | 0.0 | 1.0 |

Table 1: **Hyper-parameters used in the three-stage optimization scheme.** $w_{joint}$, $w_{reg}$, $w_{limb}$, $w_{symm}$, $w_{temporal}$, $w_\beta$, $w_{collision}$ are weights used for computing joint position, hyper-extension, limb length, limb symmetry, temporal smoothing, Gaussian mixture shape prior[2], and collision loss. Please refer to the main paper for the definition of our objective function.

leveraging human pose priors like constant limb length, joint symmetry, and temporal smoothing. The cost function is given as

$$L_{pose3D}(y) = w_l L_{limb}(y) + w_s L_{symm}(y) + w_t L_{temporal}(y) + w_i L_{reg}(y) \tag{1}$$

where $y = y_{\{1..T\}}$ and $L_{limb}$, $L_{symm}$, $L_{temporal}$, $L_{reg}$ denote constant limb length, left-right joint symmetry, temporal smoothing, and regularization losses, respectively. $wl$, $ws$, $wt$, $wi$ are scalar weights set to 1.

The definitions of $L_{limb}$, $L_{symm}$, $L_{temporal}$, $L_{reg}$ are as follows:

$$L_{limb} = \sum_{t=1}^{T-1} \|\lambda_{t+1} - \lambda_t\| \tag{2}$$

$$L_{symm} = \sum_{t=1}^{T} \left\| \lambda_t^{left} - \lambda_t^{right} \right\| \tag{3}$$

$$L_{tempotal} = \sum_{t=1}^{T-1} \|y_{t+1} - y_t\| \tag{4}$$

where $y_t$ is the 3D joint coordinates at time t, $\lambda_t$ is the limb length at time t, and $\lambda_t^{left}$ and $\lambda_t^{right}$ are the left and right limb lengths at time t.

**Mesh Fitting** We first run CLIFF[10] from all views and find the best match as our initial SMPL by comparing the limb length, keypoint coordinates, and height between all estimated SMPL and our 3D keypoints. We then fit the mesh to our 3D keypoints using the mesh fitting strategy described in the main paper. Similar to [6] and [19], the mesh is fit in a three-stage optimization scheme. Specifically, In each epoch, it first optimizes the global orientation and translation, then the body shape, followed by global orientation and translation along with body pose. The hyperparameters used in the mesh fitting process are shown in Table 1.

**Processing Time** We provide processing times for 600 timestamps using our proposed pipeline in Table 2. All experiments are conducted on an RTX2080 GPU, with average times reported over 5 runs. Note that all our annotations use the entire camera set to ensure the highest quality.

| Num. Cameras | 6 | 8 | 10 | 12 | 14 | 16 | 18 | 20 |
|---|---|---|---|---|---|---|---|---|
| MPJPE (mm) | 33.866 | 21.357 | 15.917 | 13.923 | 10.128 | 7.921 | 5.046 | 0 |
| Time | 55m | 1hr 09m | 1hr 24m | 1hr 37m | 1hr 53m | 2hr 07m | 2hr 25m | 2hr 40m |

Table 2: Average error and processing time for varying number of cameras in the Harmony4D dataset annotation pipeline.

## 2.2 Experiments

**Benchmark** We evaluate several existing monocular human mesh recovery method, including HMR2.0[5], multi-HMR[1], BUDDI[13], BEV[18], ROMP[17] and PARE[8]. To reproduce the results, please follow the instruction in : Benchmark Instruction

**Fine-tuning HMR2.0**   We fine-tuned the released checkpoint from [5] (referred to as HMR2.0b in the paper) on a single RTX 3090 GPU. We trained the model with a batch size of 16, using the AdamW optimizer. The learning rate was set to $1 \times 10^{-5}$, with $\beta1 = 0.9$, $\beta2 = 0.999$, and a weight decay of $1 \times 10^{-4}$. For each instance in our training set, we provide the ground truth SMPL parameters and 2D/3D keypoints to fine-tune HMR2.0. The 2D/3D keypoints are converted following [9] to match HMR2.0's joint order, which includes OpenPose[3] joints and those from common datasets. The finetuned checkpoint can be downloaded from : checkpoints. Please follow the benchmark instruction to reproduce the results.

## 3   Societal Impact

Harmony4D is a comprehensive dataset that provides images and detailed annotations for scenarios rich in human interactions. It can significantly advance research in areas like 3D Human Pose Estimation, Human Mesh Recovery, Analysis of Multi-Person Interaction, and the development of Virtual Reality Systems. However, this dataset also has the potential for misuse in areas like malicious surveillance, social control, and behavioral monitoring. While our dataset is not intended for these purposes, it could inadvertently contribute to them, raising ethical and societal concerns that need to be considered in future technological developments.

## 4   Datasheets for Datasets

We follow the documentation frameworks provided by Gebru et al.[4]

### 4.1   Motivation

**For what purpose was the dataset created?**

- Our dataset is created for analyzing physical interactions between humans in various real-world situations. The endeavor is to help the research community build such such system to accurately reconstruct closely interacting individuals in the wild scene.
- This dataset is related to the following 4 primary research areas, including:
    - 3D Human Pose Estimation
    - Human Mesh Recovery
    - Multi-Person Interaction
    - Virtual Reality System

**Who created the dataset (e.g., which team, research group) and on behalf of which entity (e.g., company, institution, organization)?**

- Please refer to the main paper.

**Who funded the creation of the dataset? If there is an associated grant, please provide the name of the grantor and the grant name and number.**

- The dataset is funded by CMU faculty discretionary funds.

### 4.2   Composition

**What do the instances that comprise the dataset represent (e.g., documents, photos, people, countries)? Are there multiple types of instances (e.g., movies, users, and ratings; people and interactions between them; nodes and edges)? Please provide a description.**

- The instances that comprise the dataset are the subjects interacting with each other in the scene.

**How many instances are there in total (of each type, if appropriate)?**

- There are 3.32M instances in total in our dataset.

**Does the dataset contain all possible instances or is it a sample (not necessarily random) of instances from a larger set? If the dataset is a sample, then what is the larger set? Is the sample representative of the larger set (e.g., geographic coverage)? If so, please describe how this representativeness was validated/verified. If it is not representative of the larger set, please describe why not (e.g., to cover a more diverse range of instances, because instances were withheld or unavailable).**

- Yes, it contains all possible instances.

**What data does each instance consist of? "Raw" data (e.g., unprocessed text or images) or features? In either case, please provide a description.**

- Each instance consists of a pair of an image and its corresponding annotation, including 2D bboxes, human IDs, 2D/3D human poses, and human meshes and parameters from the parametric human model.

**Is there a label or target associated with each instance? If so, please provide a description.**

- Yes, for each instance, we have the label associated with each instance.

**Is any information missing from individual instances? If so, please provide a description, explaining why this information is missing (e.g., because it was unavailable). This does not include intentionally removed information, but might include, e.g., redacted text.**

- No.

**Are relationships between individual instances made explicit (e.g., users' movie ratings, social network links)? If so, please describe how these relationships are made explicit.**

- No. Each image and its associated annotations are treated as independent instances.

**Are there recommended data splits (e.g., training, development/validation, testing)? If so, please provide a description of these splits, explaining the rationale behind them.**

- We have split the dataset. Please refer to the main paper.

**Are there any errors, sources of noise, or redundancies in the dataset? If so, please provide a description.**

- No.

**Is the dataset self-contained, or does it link to or otherwise rely on external resources (e.g., websites, tweets, other datasets)? If it links to or relies on external resources, a) are there guarantees that they will exist, and remain constant, over time; b) are there official archival versions of the complete dataset (i.e., including the external resources as they existed at the time the dataset was created); c) are there any restrictions (e.g., licenses, fees) associated with any of the external resources that might apply to a dataset consumer? Please provide descriptions of all external resources and any restrictions associated with them, as well as links or other access points, as appropriate.**

- This dataset is self-contained.

**Does the dataset contain data that might be considered confidential (e.g., data that is protected by legal privilege or by doctor–patient confidentiality, data that includes the content of individuals' non-public communications)? If so, please provide a description.**

- No.

**Does the dataset contain data that, if viewed directly, might be offensive, insulting, threatening, or might otherwise cause anxiety? If so, please describe why.**

- No.

**Does the dataset identify any subpopulations (e.g., by age, gender)? If so, please describe how these subpopulations are identified and provide a description of their respective distributions within the dataset.**

- No.

**Is it possible to identify individuals (i.e., one or more natural persons), either directly or indirectly (i.e., in combination with other data) from the dataset? If so, please describe how.**

- Participants in the dataset can be identified. They gave informed consent in accordance with IRB guidelines and were compensated for their participation.

**Does the dataset contain data that might be considered sensitive in any way (e.g., data that reveals race or ethnic origins, sexual orientations, religious beliefs, political opinions or union memberships, or locations; financial or health data; biometric or genetic data; forms of government identification, such as social security numbers; criminal history)? If so, please provide a description.**

- No.

### 4.3 Collection Process

**How was the data associated with each instance acquired? Was the data directly observable (e.g., raw text, movie ratings), reported by subjects (e.g., survey responses), or indirectly inferred/derived from other data (e.g., part-of-speech tags, model-based guesses for age or language)? If the data was reported by subjects or indirectly inferred/derived from other data, was the data validated/verified? If so, please describe how.**

- We used our multi-view capture system.

**What mechanisms or procedures were used to collect the data (e.g., hardware apparatuses or sensors, manual human curation, software programs, software APIs)? How were these mechanisms or procedures validated?**

- Please refer to the main paper.

**If the dataset is a sample from a larger set, what was the sampling strategy (e.g., deterministic, probabilistic with specific sampling probabilities)?**

- This dataset is not a sample.

**Who was involved in the data collection process (e.g., students, crowdworkers, contractors) and how were they compensated (e.g., how much were crowdworkers paid)?**

- People in the collection process includes students, professional dancers, and MMA fighters. Participants receive 25 and 50 US dollar per hour for Student participants and Professional participants, respectively.

**Over what timeframe was the data collected? Does this timeframe match the creation timeframe of the data associated with the instances (e.g., recent crawl of old news articles)? If not, please describe the timeframe in which the data associated with the instances was created.**

- The data is collected on September, 2023. It matches the creation timeframe of the data assiciated with the instances.

**Were any ethical review processes conducted (e.g., by an institutional review board)? If so, please provide a description of these review processes, including the outcomes, as well as a link or other access point to any supporting documentation.**

- No.

**Did you collect the data from the individuals in question directly, or obtain it via third parties or other sources (e.g., websites)?**

- We collect the data from the individuals in question directly.

**Were the individuals in question notified about the data collection? If so, please describe (or show with screenshots or other information) how notice was provided, and provide a link or other access point to, or otherwise reproduce, the exact language of the notification itself.**

- Yes, all participants are briefed the project purpose. link. Below is the exact language of the notification:

  In connection with the Harmony4D project, we will attempt to collect and record a 2-dimensional ("2D") and 3-dimensional ("3D") dataset of persons performing close human interaction activities through ARIA glasses worn by study participants and/or stationary GoPro cameras mounted around the study participants.

**Did the individuals in question consent to the collection and use of their data? If so, please describe (or show with screenshots or other information) how consent was requested and provided, and provide a link or other access point to, or otherwise reproduce, the exact language to which the individuals consented.**

- Yes, all participant consent to the collection and use of their data. Consent was provided from link.

**If consent was obtained, were the consenting individuals provided with a mechanism to revoke their consent in the future or for certain uses? If so, please provide a description, as well as a link or other access point to the mechanism (if appropriate).**

- Yes.

**Has an analysis of the potential impact of the dataset and its use on data subjects (e.g., a data protection impact analysis) been conducted? If so, please provide a description of this analysis, including the outcomes, as well as a link or other access point to any supporting documentation.**

- No.

### 4.4 Preprocessing/cleaning/labeling

**Was any preprocessing/cleaning/labeling of the data done (e.g., discretization or bucketing, tokenization, part-of-speech tagging, SIFT feature extraction, removal of instances, processing of missing values)? If so, please provide a description. If not, you may skip the remaining questions in this section.**

- No.

### 4.5 Uses

**Has the dataset been used for any tasks already? If so, please provide a description.**

- Yes, this dataset has been used for evaluating SOTA mesh recovery method.

**Is there a repository that links to any or all papers or systems that use the dataset? If so, please provide a link or other access point.**

- No.

**What (other) tasks could the dataset be used for?**

- This dataset can be used, but not limited to:

- 3D Pose Etimation
- Development of Virtual Reality System
- Analysis of Human Interactions
- Human Mesh Recovery

**Is there anything about the composition of the dataset or the way it was collected and preprocessed/cleaned/labeled that might impact future uses? For example, is there anything that a dataset consumer might need to know to avoid uses that could result in unfair treatment of individuals or groups (e.g., stereotyping, quality of service issues) or other risks or harms (e.g., legal risks, financial harms)? If so, please provide a description. Is there anything a dataset consumer could do to mitigate these risks or harms?**

- No.

**Are there tasks for which the dataset should not be used? If so, please provide a description.**

- No.

## 4.6 Distribution

**Will the dataset be distributed to third parties outside of the entity (e.g., company, institution, organization) on behalf of which the dataset was created? If so, please provide a description.**

- No.

**How will the dataset will be distributed (e.g., tarball on website, API, GitHub)? Does the dataset have a digital object identifier (DOI)?**

- The dataset will be hosted on Google Drive for distribution.

**When will the dataset be distributed?**

- We planned to release the dataset at the end of September.

**Will the dataset be distributed under a copyright or other intellectual property (IP) license, and/or under applicable terms of use (ToU)? If so, please describe this license and/or ToU, and provide a link or other access point to, or otherwise reproduce, any relevant licensing terms or ToU, as well as any fees associated with these restrictions.**

- Yes. For research Purpose.

**Have any third parties imposed IP-based or other restrictions on the data associated with the instances? If so, please describe these restrictions, and provide a link or other access point to, or otherwise reproduce, any relevant licensing terms, as well as any fees associated with these restrictions.**

- No.

**Do any export controls or other regulatory restrictions apply to the dataset or to individual instances? If so, please describe these restrictions, and provide a link or other access point to, or otherwise reproduce, any supporting documentation.**

- No.

## 4.7 Maintenance

**Who will be supporting/hosting/maintaining the dataset?**

- The authors of this paper will be maintaining the dataset.

**How can the owner/curator/manager of the dataset be contacted (e.g., email address)?**

- We will provide the contact information after releasing the dataset.

**Is there an erratum? If so, please provide a link or other access point.**

- No.

**Will the dataset be updated (e.g., to correct labeling errors, add new instances, delete instances)? If so, please describe how often, by whom, and how updates will be communicated to dataset consumers (e.g., mailing list, GitHub)?**

- Yes. The authors of this paper will update it.

**If the dataset relates to people, are there applicable limits on the retention of the data associated with the instances (e.g., were the individuals in question told that their data would be retained for a fixed period of time and then deleted)? If so, please describe these limits and explain how they will be enforced.**

- No.

**Will older versions of the dataset continue to be supported/hosted/maintained? If so, please describe how. If not, please describe how its obsolescence will be communicated to dataset consumers.**

- No. We simply add more instances to the dataset.

**If others want to extend/augment/build on/contribute to the dataset, is there a mechanism for them to do so? If so, please provide a description. Will these contributions be validated/verified? If so, please describe how. If not, why not? Is there a process for communicating/distributing these contributions to dataset consumers? If so, please provide a description.**

- People who wants to contribute/extend the dataset. Please contact the author of this paper.

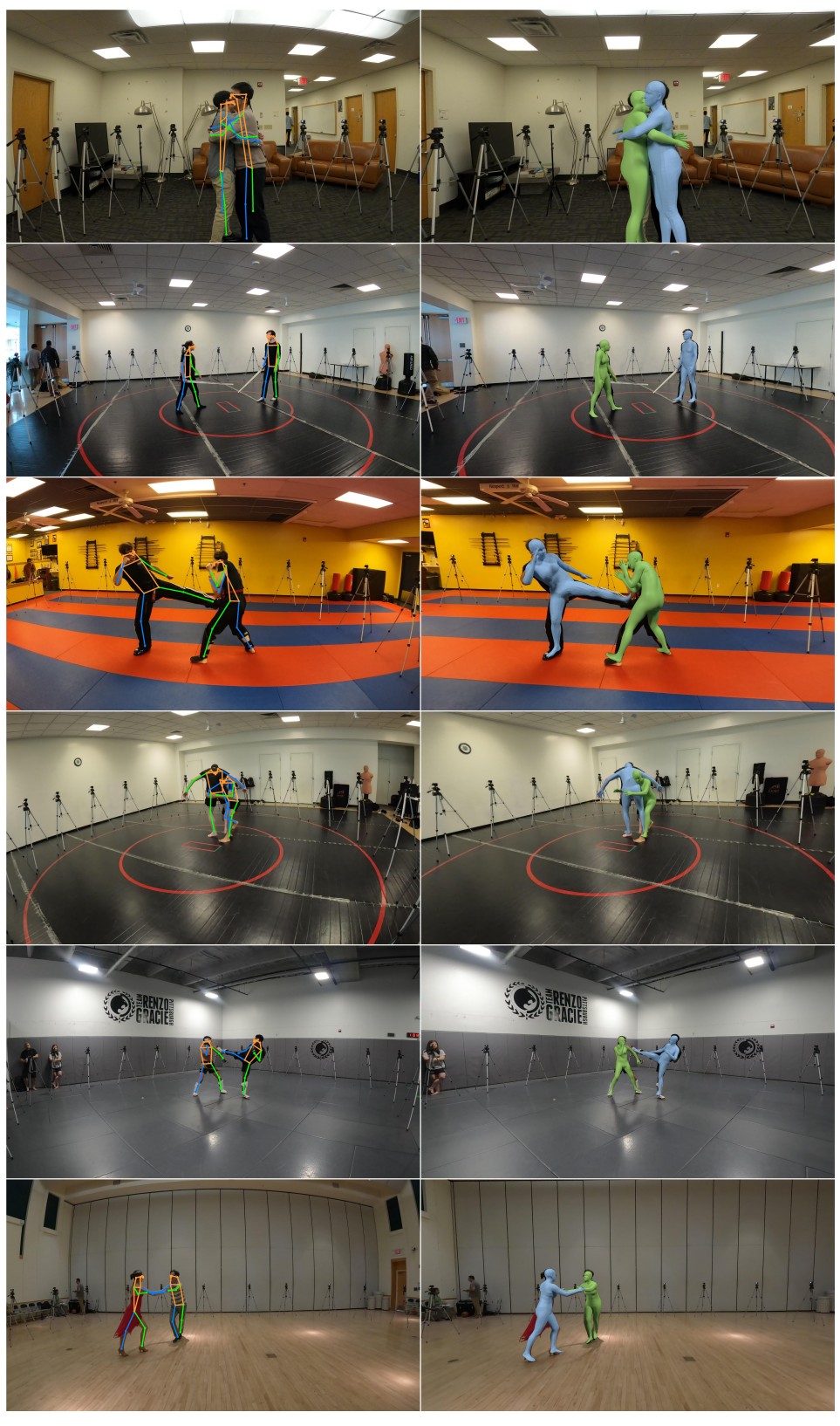

Figure 2: **Examples from Harmony4D Dataset.** The left side shows the 3D pose annotation and the right side shows the human meshes.

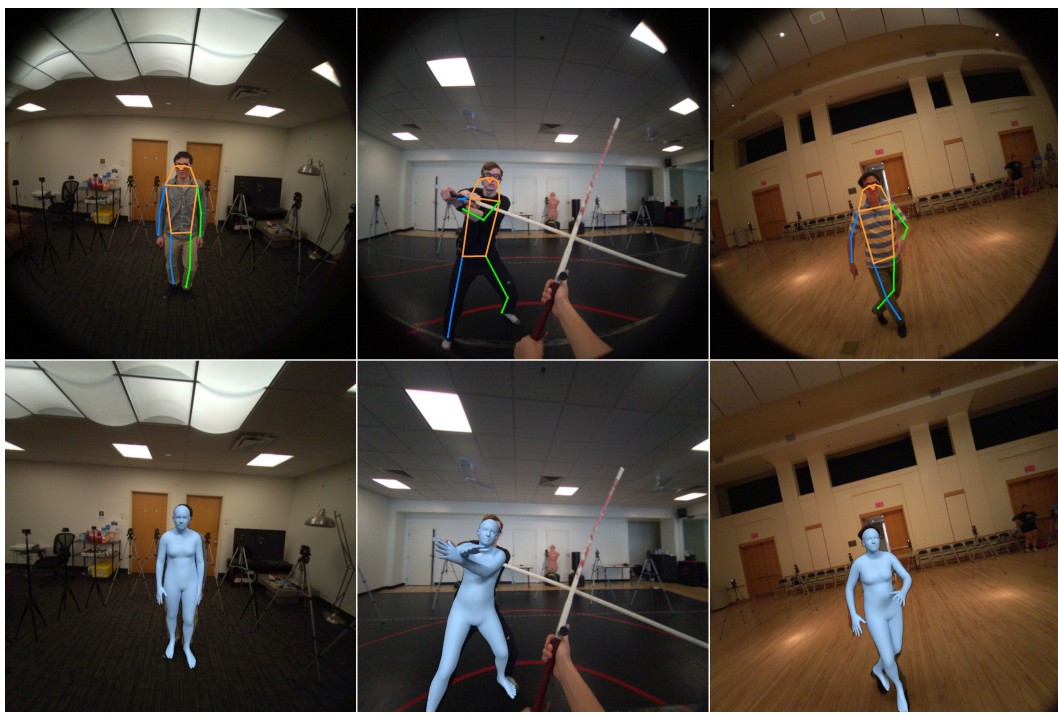

Figure 3: **Example of our annotations on the ego-centric images for hugging, fencing, and dancing sequences.** The ego-centric image is captured by Meta's Aria glasses. The first row shows the 3D pose, and the second row displays the human mesh.