# OpenReview forum: "Harmony4D: A Video Dataset for In-The-Wild Close Human Interactions"
_NeurIPS.cc/2024/Datasets_and_Benchmarks_Track — NeurIPS 2024 Track Datasets and Benchmarks Poster_

### Official Review · Reviewer_h7vn · 2024-07-23
**A timely dataset for modeling human-human interactions**

**Rating:** 7
**Confidence:** 4
**Clarity:** Yes the paper is clearly written.

**Review:**

This paper introduces a large-scale multi-view captured video dataset of in-the-wild human interactions. The scale and in-the-wild nature of the dataset is timely and has significance for the advancement of modeling human-human interaction.

## Pros
- The proposed dataset has significance on the human-human interaction domains. Existing datasets lack the diversity of interactions and environments. The proposed dataset diversified the types of interactions and environments and increased the scale of the dataset.
- The authors demonstrate the limitation of existing datasets by showing the SOTA methods trained on them fail on the proposed dataset. At the same time, the authors show that training on the proposed dataset leads to large improvement.
- The proposed pipeline for precise mesh registration is convincing.

## Cons
- The necessity of ARIA glasses is not well-demonstrated. What benefits do ARIA glasses provide in this capture setup?
- Fitting SMPL-X(at least SMPL-H) should have been much better. The authors even had Ego-view captures which can provide precise hand information. The previous work CHI3D provides SMPL-X annotations.
- Interpenetration loss is not novel which was widely used in previous literature on human-object interaction. The authors should have included a reference for it.  (e.g., Xu, Sirui, et al. "Interdiff: Generating 3d human-object interactions with physics-informed diffusion.", ICCV 2023.)
- Ablation study is not very informative. The fact that increasing the number of cameras leads to the precise result is quite obvious. I'm not convinced with the fact that the 16 cameras offer a fair trade-off between accuracy and processing speed is interesting. Does decreasing 4 cameras lead to significant improvement of processing speed? There lacks information about the processing speed.

**Strengths:**

Despite the cons mentioned, the dataset has significance for modeling 3D human-human datasets.
This dataset includes challenging situations that previous datasets haven't covered. This will impose a new challenge on the field and be a good source of improvement for the monocular human mesh recovery method.
The provided details of the mesh annotation process will be useful for capturing a wide range of human interactions.

**Additional Feedback:**

Minor comments
- Please avoid using the reference as nouns., e.g., "following [39]". The sentence should be complete without the reference.
- The reference for figures are not consistent, e.g., Fig.X, Figure X.
- The details on vertices in contact seems missing. Above 4.2, there are words referring the readers to the supplementary materials for the vertices in contact, I cannot find the info in the supp.

**Correctness:**

Yes, the dataset construction process is sound and the capture process is well-described.

**Documentation:**

Yes, they include sufficient detail on data collection and related info.

**Ethics:**

No, I don't suspect any ethical concerns.

**Limitations:**

Yes, the authors mentioned the limitations of their work. They provided proper information to participants for data collection to avoid potential issues.

**Opportunities For Improvement:**

- I believe SMPL-X fitting on the captured dataset will boost the usefulness of this dataset.
- The usefulness of ego-view capture is not well covered. Demonstrations of the usefulness of the ego-view capture will justify the capture setup of this dataset.

**Relation To Prior Work:**

They discussed the difference between previous datasets and the proposed dataset.

**Summary And Contributions:**

This paper introduces a multi-view video dataset of in-the-wild human-human interactions, e.g., wrestling, dancing, etc. The authors propose a novel tracking algorithm to get precise 3D mesh annotations without specialized markers. While tracking 3D pose with the Kalman filter, they get precise instance masks of interacting humans using the projected 3D keypoints, then get more precise 2D pose using the instance segmentation masks. To mitigate the inter/intra-penetration of human meshes, they fit the SMPL mesh with collision loss. They show that SOTA monocular mesh prediction methods fail on their dataset, demonstrating that the existing training dataset doesn't cover the challenging interactions. By fine-tuning the HMR2.0 on the proposed dataset, they see performance improvement.

---

> ### Author Rebuttal · Authors · 2024-08-17
>
> Thank you for your feedback and acknowledging the technical contributions of Harmony4D.
>
> **Benefits of Aria Glasses**:
> - As mentioned in Sec. 3 (Data Collection), Aria glasses are optionally included in our dataset to support research in ego-pose estimation for challenging multi-human interaction sequences.
> - Our proposed method described in Sec. 3 does not rely on egocentric views and only assumes 3D pose initializations for the pre-contact stage. This initialization is currently done manually or automatically using the 3D camera centers of the Aria glasses (for more details, we refer to EgoHumans, ICCV 2023).
>
> **Fitting SMPL-X**:
> - Thank you for the suggestion. Fitting an expressive body model like SMPL-X (face + body + feet + hands) is indeed challenging for sequences with severe occlusion, which is the focus of this work.
> - Recovering hand and face keypoints solely from GoPros is challenging due to limited resolution. We demonstrate the ego-view videos on our [project page](https://jyuntins.github.io/harmony4d/) to show that hands are often out-of-sight of the subject when performing dynamic activities.
> - Following CHI3D, manually annotating hand and face keypoints on images is a promising direction to recover an expressive whole body model. In our context, this is significantly expensive given the scale of the dataset. Nevertheless, we are committed to explore this as an extension of our work.
>
> **Missing Interpenetration Loss Citation**:
> - We do not claim novelty in the formulation of the interpenetration loss and will update the text to clarify this.
> - We have added the suggested citation to the references.
>
> **Missing Processing Speeds**:
> - As requested, we provide processing times for 600 timestamps using our proposed pipeline.
> - All experiments are conducted on an RTX2080 GPU, with average times reported over 5 runs.
> - Note that all our annotations use the entire camera set to ensure the highest quality.
>
> | Num. Cameras | MPJPE (mm) | Time |
> | :---------------- | :-------: | ---: |
> | 20                |     -     | 41m  |
> | 18                |  5.046    | 37m  |
> | 16                |  7.921    | 32m  |
> | 14                | 10.128    | 29m  |
> | 12                | 13.923    | 25m  |
> | 10                | 15.917    | 22m  |
> | 8                 | 21.357    | 18m  |
> | 6                 | 33.866    | 15m  |
>
>
> **Minor**:
> - We have updated the manuscript to remove citations as nouns and use consistent terminology for figures.
> - In Sec. 4.2, we imply reference to the supplemental for additional implementation details. Please refer to the [project page](https://jyuntins.github.io/harmony4d/) for contact vertex videos.

---

> > ### Comment · Reviewer_h7vn · 2024-08-27
> >
> > I appreciate the anwsers for all the things I pointed out. I understand that SMPL-X fitting is quite demanding considering the resolutions of the camera used, occlusion, etc. Processing speed report is reasonable and better supports the claim.
> > I believe this dataset is meaningful for the vision community in the current state and authors made proper justification for the things I pointed out. Therefore I raise my score to 7. I hope the authors make more progress in this direction.

---

> > > ### Author Response · Authors · 2024-09-04
> > >
> > > Thank you for reviewing our work and raising the score. We appreciate the encouraging feedback!

---

### Official Review · Reviewer_nxLE · 2024-07-28

**Rating:** 6
**Confidence:** 4
**Correctness:** YES
**Clarity:** Fair

**Review:**

See strengths and Improvements

**Strengths:**

- First dataset capturing in-the-wild dynamic activities and contact interactions.
- Evaluations reveal that existing state-of-the-art methods underperform on proposed datasets.

**Additional Feedback:**

See above

**Documentation:**

Good

**Limitations:**

YES

**Opportunities For Improvement:**

- Limited number of subjects, with only 24 compared to datasets [86,79] which have more, and only 5 scenes which may seem restricted.
- Potential inaccuracies in calibration using structure from motion (SfM).
- Absence of video demonstrations to visualize and check temporal consistency.
- Inconsistency in Figure 8, where the ground truth mesh has shading while others do not.

**Relation To Prior Work:**

Fair

**Summary And Contributions:**

Understanding human interaction is vital for creating realistic multi-human virtual reality systems, but this area lacks large-scale datasets. Harmony4D addresses this by providing a multi-view video dataset of diverse, in-the-wild human interactions with extensive annotations. The dataset improves performance in modeling close interactions and occlusions, demonstrated through fine-tuning a pre-trained model.

---

> ### Author Rebuttal · Authors · 2024-08-16
>
> Thank you for your valuable comments.\
> We appreciate the opportunity to address these concerns, which are listed below.
>
> **Limited number of subjects and scenes compared to MultiHuman and Hi4D**:
> - Based on our correspondence with the authors of MultiHuman[86], we confirmed that the dataset consists of only 8 subjects. The reported number of 284 subjects in [86] is erroneous and will be corrected.
> - Compared to Hi4D[79] dataset’s 40 subjects, our 24 participants include professional athletes, martial artists, and dancers, representing a more socially diverse set.
> - Unlike existing datasets captured at a single location, Harmony4D encompasses multiple natural environments specific to each activity.
>
> Importantly, we will release both the dataset and annotation processing code, along with steps to replicate our capture setup, enabling researchers to collect their in-the-wild datasets at diverse locations.
>
> **Calibration Inaccuracies**:
> - As mentioned in Sec. 4.1, we manually inspect and rectify the 3D poses for each time step by reprojecting them to all camera views.
> - Before this, we also visualize the GoPro camera centers in all views to verify the accuracy of the structure-from-motion calibration.
> - Sequences with low-quality calibration are either re-calibrated or discarded, ensuring only high-quality data is included in our final dataset.
>
> **Missing Video Demonstrations**:
> - Please refer to the supplementary for the Harmony4D project page: https://jyuntins.github.io/harmony4d/.
> - We present multiple activities along with their corresponding single and multi-view annotations, highlighting the temporal consistency of our method.
>
> **Updates to Figure 8**:\
> Thank you for pointing this out. We have updated Figure 8 to use a consistent renderer for ground truth and all methods.

---

> > ### Author Response · Authors · 2024-09-04
> >
> > Hello,
> > Please let us know if you would like any further clarifications.
> > Thank you.

---

### Official Review · Reviewer_ebSy · 2024-07-29
**review of submission 546**

**Rating:** 6
**Confidence:** 5
**Clarity:** It is well-written.

**Review:**

The developed dataset and annotation pipeline require significant efforts and experiments. It is non-trivial to build such a large-scale dataset. This dataset can be a good addition to the 3D human pose field, providing more challenging and rich training data for improving performance in the close-contact scenario.

**Strengths:**

- This paper is currently the largest close-contact human dataset at this point.
- The authors clearly describe their data collection setup and the proposed human mesh tracking system.
- The proposed pre-contact and post-contact stages are generally sound and reasonable. Leveraging a human motion model to reason the occluded keypoints is also technically valid.
- The authors also provide performance benchmarking on the proposed Harmony4D test set. This gives a clear understanding of the performance of SOTA models, and fine-tuning HMR2.0 gives notable performance improvements.

**Additional Feedback:**

N/A

**Correctness:**

The evaluation methods and experiment design are appropriate. The data capture setup and pre/post contact processing are technically valid.

**Documentation:**

The papers present sufficient details. However, in my opinion, it may not be easy to reproduce the data capture setup due to camera specs and hardware requirements.

**Limitations:**

I don't see potential negative societal impact. Meanwhile, the authors discussed sufficient amounts of limitations of the proposed dataset.

**Opportunities For Improvement:**

- The collected data remains restricted in many aspects. For environments, there are only indoor scenarios based on the diagram, figure and data statistics. However, in Sec 2, the authors claimed that Harmony4D includes both indoor and outdoor environments. The statement is confusing. Moreover, for the considered activities, there are only 6 available, which is also limited.
- It would be helpful to discuss the common failure cases in the proposed dataset, and provide occlusion-related insights and potential directions, which will motivate and inspire future researchers.

**Relation To Prior Work:**

The authors provide sufficient discussions and comparisons with the related work.

**Summary And Contributions:**

The key contribution is a large-scale multi-view human dataset, focusing on close-contact human interaction.

To build such dataset, a scalable annotation pipeline is needed, and with the focus of close-contact interaction, one needs to detect and assign the partially/completed occluded keypoints to the correct person identities. To this end, this paper presents a post-contact processing and borrows existing 3D human pose forecasting model with KF filter to help track the keypoints.

---

> ### Author Rebuttal · Authors · 2024-08-16
>
> Thank you for recognizing the contributions of our paper and for providing insightful comments.
>
> **Limited Collected Data**:
> - In contrast to existing datasets like Hi4D and CHI3D, the activities captured in Harmony4D are highly dynamic and unchoreographed. The participating subjects received no prior instructions on how to interact, resulting in completely natural motion.
> - Furthermore, as mentioned in the strengths section of the review, our work currently represents the largest in-the-wild close-contact human dataset available consisting of over 20 calibrated and synchronized camera views with 1.66M images and 3.32M instance annotations.
> - Unlike prior methods, our activities are not limited to a singular capture setup.
>
> Inspired by the feedback, following Hi4D (Fig. 4), we showcase the contact coverage and contact frequency in our dataset in the attached pdf. The activities in our dataset ensure greater than 95% body coverage due to challenging activities like dancing, wrestling and martial arts showcasing the diversity of our dataset.
>
> **Environments (Indoor/Outdoor)**:
> - Harmony4D is primarily designed for in-the-wild conditions and our proposed setup is applicable to both indoor as well as outdoor conditions.
> - We will revise Sec. 2 to replace _indoor/outdoor_ with _in-the-wild_ environments to be consistent with activities collected in the dataset.
>
> **Discussion on Failure Cases + Potential Future Directions**:
> - We describe failure cases in the Limitations section of the main paper and will expand on this with more details and examples in the camera-ready version.
> - Potential future directions include adding more keypoints to the base skeleton, such as face, fingers, and feet. Additionally, we aim to use Harmony4D video annotations to train video-based mesh regressors capable of reasoning about occlusion and human-human interactions over longer time horizons. We will modify the manuscript to include this discussion.
>
> **Occlusion-Related Insights**:
> - As requested, we provide occlusion-based performance analysis of baselines and finetuned methods on the Harmony4D dataset in Table. 1 of the attached pdf.
> - We classify _test_ set images into low, medium and high occlusion categories based on the ground-truth bounding box IoU thresholds.
> - We observe that as the severity of occlusion increases, the prediction accuracy of baseline methods decreases in multi-human contact scenarios. For instance, Multi-HMR achieves an MPJPE of 117.3 mm on the high occlusion set.
> - Interestingly, finetuned HMR2.0 consistently outperforms baselines across all occlusion levels, with performance gain increasing with occlusion.

---

> > ### Comment · Reviewer_ebSy · 2024-08-31
> > **comments**
> >
> > Thanks for providing the detailed feedback. I would suggest adding the discussions to the final paper or appendix.
> > I would recommend accepting this paper.

---

> > > ### Author Response · Authors · 2024-09-04
> > >
> > > Moving the discussions to appendix. Thank you for reviewing this work.

---

### Author Rebuttal · Authors · 2024-08-17

**General Response**

We sincerely thank all the reviewers (R1-ebSy, R2-nxLE, R3-h7vn) for their insightful suggestions and positive feedback. We are greatly encouraged by the unanimous pre-rebuttal acceptance.

More specifically, the reviewers have appreciated the great importance of a large-scale in-the-wild dataset for multi-human contact interactions. R1-ebSy mentioned our technical contributions and the significant efforts required, R2-nxLE highlighted its novelty in capturing dynamic activities and contact interactions, and R3-h7vn emphasized the dataset's coverage of previously uncaptured challenging situations.

Thank you again. As the reviewers raised distinct questions, we have provided individual responses to each below.

---

### Decision · Program_Chairs · 2024-09-26

**Decision:**

Accept (Poster)

**Comment:**

This paper uses a flexible multi-view capture system to record dynamic activities and provides annotations for human detection, tracking, 2D/3D pose estimation, and mesh recovery for closely interacting subjects. It uses a novel markerless algorithm to track 3D human poses in severe occlusion and close interaction to obtain annotations with minimal manual intervention. The proposed Harmony4D dataset consists of 1.66 million images and 3.32 million human instances from more than 20 synchronized cameras with 208 video sequences spanning diverse environments and 24 unique subjects.

All of the three reviewers support the acceptance of this paper. The paper is overall well organized and the dataset is helpful for human-interaction analysis. The rebuttal partially addresses some of the reviewers' concerns. AC agrees with reviewers to accept this paper.